# Joule heating in bad and slow metals

Paolo Glorioso[1] and Sean A. Hartnoll[2]

**1** Department of Physics, Stanford University, Stanford, CA 94305-4060, USA
**2** Department of Applied Mathematics and Theoretical Physics,
University of Cambridge, Cambridge CB3 0WA, UK

## Abstract

Heat supplied to a metal is absorbed by the electrons and then transferred to the lattice. In conventional metals energy is released to the lattice by phonons emitted from the Lindhard continuum. However in a 'bad' metal, with short mean free path, the low energy Lindhard continuum is destroyed. To describe energy transfer to the lattice in these cases we obtain a general Kubo formula for the energy relaxation rate in terms of the electronic density spectral weight $\mathrm{Im}\, G_{nn}^R(\omega_k, k)$ evaluated on the phonon dispersion $\omega_k$. We apply our Kubo formula to the high temperature Hubbard model, using recent data from quantum Monte Carlo and experiments in ultracold atoms to characterize $\mathrm{Im}\, G_{nn}^R(\omega_k, k)$. We furthermore use recent data from electron energy-loss spectroscopy to estimate the energy relaxation rate of the cuprate strange metal to a high energy optical phonon. As a second, distinct, application of our formalism we consider 'slow' metals. These are defined to have Fermi velocity less than the sound velocity, so that particle-hole pairs are kinematically unable to emit phonons. We obtain an expression for the energy relaxation rate of a slow metal in terms of the optical conductivity.

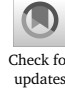
## Contents



# 1 Joule heating

The dc electrical conductivity $\sigma$ of a closed quantum system is defined via the linear response of the system to a uniform electric field $j = \sigma E$. However, beyond linear order the electric field causes Joule heating. Energy density $\varepsilon$ is pumped into the system at a rate $\dot{\varepsilon} = j \cdot E = \sigma E^2$. If the system is truly closed then this energy has no way to dissipate, the system will heat up and a stationary state is not possible, rendering linear response ill-defined.

Real-world condensed matter systems are not closed and can dissipate energy to the environment. An important energy sink are the neutral lattice vibrations. The transfer of energy from the electronic degrees of freedom to the lattice has been long studied theoretically [1–3]. More recently, the timescale $\tau$ over which the electronic degrees of freedom release energy to the lattice has been probed using ultrafast spectroscopy [4]. In these experiments, energy supplied to the material naturally heats up the electrons relative to the lattice, as the electronic degrees of freedom have a lower specific heat. The subsequent cooling of the electronic subsystem, as energy is transferred to the lattice, can be followed in detail using time- and angle-resolved photoemission (tr-ARPES) [5–8].

In the experiments described above, it is observed that the electronic subsystem thermalizes on a much faster timescale than $\tau$. This means that throughout the cooling process the electrons have a well-defined, time-dependent temperature $T(t)$. Such rapid redistribution of energy among electronic degrees of freedom requires electronic interactions. However, in some tension with this fact, the established theory of energy relaxation to the lattice describes phonons emitted by the Lindhard continuum of non-interacting particle-hole pairs [1, 3]. In conventional metals, as we explain below, this tension is resolved by the fact that most phonons are emitted by low energy fluctuations of particle-hole pairs on inverse momentum scales shorter than the electronic mean free path. Such fluctuations are not strongly affected by interactions.

In non-Fermi liquids, however, strong electronic interactions can lead to the absence of coherent particle-hole excitations at any momenta. While many theoretical and experimental results support this statement, perhaps the most directly relevant for our purposes are recent measurements of the dynamic charge susceptibility using momentum-resolved electron energy-loss spectroscopy (M-EELS) in the non-Fermi liquid regime of a cuprate [9, 10]. These have revealed the conspicuous absence of a well-defined Lindhard continuum in the charge density spectral function $\text{Im}\,G_{nn}^R(\omega, k)$. If there is indeed no Lindhard continuum then the

established theory of energy loss to the lattice is clearly inapplicable.

In this work we obtain a Kubo formula for the timescale $\tau$ that does not start from non-interacting electrons. Instead, our approach will make a virtue of the fact that electronic thermalization is faster than equilibration with the lattice. Energy transfer to the lattice can therefore be treated as a perturbative process that occurs in an electronically thermalized medium. This leads to our formula (14) below for $\tau$ in terms of the equilibrium electron density spectral function $\text{Im}\, G^R_{nn}(\omega, k)$ integrated along the phonon dispersion $\omega = \omega_k$. This particular spectral function appears because we will consider lattice vibrations that couple to the charge density via a deformation potential.[1] It is the same object that can be extracted from M-EELS measurements and, needless to say, is among the most basic quantities characterizing an electronic medium. To illustrate the application of our Kubo formula to an incoherent metal, we will use the recent M-EELS data to estimate the rate of energy dissipation from the cuprate strange metal into a high-frequency optical phonon.

In conventional metals with a long mean free path our framework will be shown to precisely recover the classic results [1, 3]. More generally, our formula can be used with or without coherent particle-hole excitations to quantitatively obtain the relaxation rate $1/\tau$ once the phonon dispersions and $\text{Im}\, G^R_{nn}(\omega, k)$ are known from experiment or numerics. This connection can be tested in spectroscopic pump-probe experiments and also in measurements of non-linear current response, as we describe below.

Bad metals [11–13] are especially well-suited to our Kubo formula because the very short mean free path means that the entire low energy Lindhard continuum is destroyed, and $\text{Im}\, G^R_{nn}(\omega, k)$ instead acquires a universal diffusive form. We obtain a simple expression for the relaxation rate $1/\tau$ in this limit, which we proceed to evaluate using recent measurements of diffusion in an ultracold atom simulation of the Hubbard model at high temperatures [14]. These high temperatures are comparable to those arising in pump-probe experiments in which the electrons become very hot.

A second, distinct, application of our formula is to 'slow' metals, where the Fermi velocity is small compared to the sound speed. In this case the phonon dispersion is outside of the Lindhard continuum. Electronic interactions are therefore essential to broaden the charge density spectral weight to low momenta, allowing phonons to be emitted. We obtain the decay rate $1/\tau$ using our Kubo formula together with a low momentum spectral weight consistent with a Drude conductivity. Small Fermi velocities could in principle arise in semi-metals, in doped semiconductors, or if an effective mass divergence strongly renormalizes the Fermi velocity down, as occurs close to a quantum critical point.

## 2 A Kubo formula for energy dissipation

### 2.1 Stationary state current with an applied field

Consider a Hamiltonian for electrons and phonons (lattice vibrations) of the form

$$H = H_{\text{e}} + H_{\text{e-ph}} + H_{\text{ph}}. \tag{1}$$

We will obtain the rate at which energy is transferred from the electronic degrees of freedom to the lattice, working perturbatively in the electron-phonon interaction $H_{\text{e-ph}}$. The electronic Hamiltonian $H_{\text{e}}$ can be strongly correlated. We consider the case where the electrons and phonons are only slightly perturbed away from equilibrium.

---

[1] We will not consider other couplings such as would arise for bond phonons, for example. However, our general framework can be applied to any form of coupling between the lattice and electronic subsystems.

Let $\delta\varepsilon$ be the difference of the energy density of the electronic system from the value set by thermal equilibrium with the lattice. The lattice is assumed to be kept at a constant temperature by rapid equilibration with the external environment. The energy density then obeys the conservation equation

$$\partial_t \delta\varepsilon + \nabla \cdot j_\varepsilon = E \cdot j - \frac{1}{\tau}\delta\varepsilon\,, \tag{2}$$

In this section we will obtain a Kubo formula for the energy relaxation rate $1/\tau$. Equation (2) is precise if the electronic system locally equilibrates itself on timescales faster than $\tau$, because in that case all non-conserved local quantities will have already decayed and one need only keep track of the energy perturbation $\delta\varepsilon$. More generally, (2) may be a reasonable 'relaxation time approximation' for energy dissipation. The setup is illustrated in Fig. 1

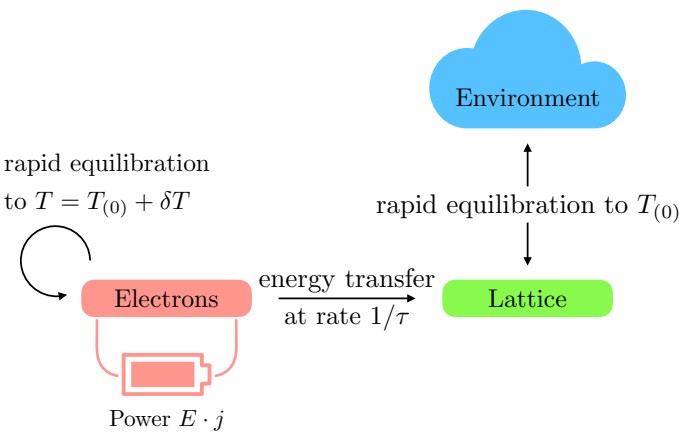

Figure 1: Power is supplied to the electrons by driving a current. The electrons rapidly thermalize to a temperature $\delta T$ above the temperature of the thermalized lattice and environment. Heat is consequently transferred to the lattice, at a rate $1/\tau$.

Firstly, we will illustrate the physics of the timescale $\tau$. Time-independent and spatially uniform solutions to (2) are obtained by balancing the two terms on the right hand side. Thus we see that the applied field causes a shift in the electronic temperature away from the value $T_{(0)}$ set by the lattice degrees of freedom. That is, $T = T_{(0)} + \delta T$ with

$$\delta T = \frac{\delta\varepsilon}{c} = \frac{\tau}{c} E \cdot j = \frac{\tau\sigma_{(0)}}{c}E^2 + \cdots. \tag{3}$$

Here $c$ is the electronic specific heat and in the final equality we have worked to leading order in small $\delta T$ so that $j = \sigma(T_{(0)})E \equiv \sigma_{(0)}E$. We will assume spatial isotropy for simplicity. The shift in temperature $\delta T$ is seen in (3) to depend on the applied field, and therefore leads to non-Ohmic conduction. Accounting for this shift

$$j^i = \sigma(T)E^i + \cdots = \sigma_{(0)}E^i + \frac{\tau\sigma_{(0)}\sigma'_{(0)}}{c}E^2 E^i + \cdots. \tag{4}$$

Here we expanded $\sigma(T) = \sigma_{(0)} + \sigma'_{(0)}\delta T + \cdots$, where $\sigma'_{(0)}$ is the temperature derivative of the conductivity. The $\cdots$ terms in (4) include other sources of nonlinear current response. The correction shown in (4) will dominate other effects as $\tau$ becomes large. This is when the coupling to the lattice is weak.

Equation (4) shows that while there is no steady state as $\tau \to \infty$, as anticipated, for any finite $\tau$ the dc conductivity is given by $\sigma_{(0)}$ for sufficiently small electric fields. More

generally, (4) shows that the timescale $\tau$ determines the electric field (or, equivalently, the critical current) above which conduction becomes non-Ohmic. From balancing the first two terms on the right hand side of (4), the critical current is

$$j_{\text{non-Ohm}} = \sigma_{(0)} \sqrt{\frac{c}{\tau |\sigma'_{(0)}|}}.$$ (5)

Note that in a metal $\sigma'_{(0)} < 0$, hence the absolute value in the above equation. We will return to this expression once we have a better handle on $\tau$. Once conduction is non-Ohmic, for $j > j_{\text{non-Ohm}}$, higher order corrections to (4) are likely to be important.

## 2.2 Energy relaxation rate due to phonons

From the conservation law (2), the total electron energy $E = \delta \varepsilon_{k=0} = H_{\text{e}} - \langle H_{\text{e}} \rangle_{(0)}$ has retarded Green's function [15]

$$G^R_{EE}(\omega) = \frac{cT/\tau}{i\omega - 1/\tau}.$$ (6)

Using (6), the energy relaxation rate is seen to be

$$\frac{cT}{\tau} = -\lim_{\omega \to 0} \lim_{\omega\tau \gg 1} \omega \operatorname{Im} G^R_{EE}(\omega)$$ (7)

$$= -\lim_{\omega \to 0} \lim_{\omega\tau \gg 1} \frac{1}{\omega} \operatorname{Im} G^R_{\dot{E}\dot{E}}(\omega).$$ (8)

The order of limits is important. We need to take $\omega$ small compared with electronic equilibration times in order to be in the low energy regime universally described by (6), while $\omega$ must be large compared to $1/\tau$ to pick out the numerator of (6). These two conditions are compatible if, as we have already described above, $\tau$ is long compared to purely electronic timescales. We noted in the introduction that experiments in strongly correlated materials indicate that the electrons indeed self-thermalize faster than the combined electron-phonon system. This will allow us to treat the electron-phonon interaction as a perturbation of the thermalized electronic dynamics. In the absence of such a hierarchy of timescales, the full coupled electron-phonon problem would need to be solved, which is beyond the methods we develop below. Our approach will also implicitly assume that the backreaction of phonons on the electronic sector, which will be neglected, does not lead to any qualitatively important effects, as we discuss further in section 4 below.

The second line (8) is a commonly used step in the memory function approach to relaxation rates [16]. It follows from the general relation $G^R_{\dot{A}B}(\omega) = -i\omega G^R_{AB}(\omega) + \chi_{\dot{A}B}$, together with the fact that the susceptibility $\chi_{\dot{E}E} = 0$. The reason to make the step to (8) is that $\dot{E}$ is proportional, as an operator, to the electron-phonon interaction. This is because the total electron energy is conserved without the coupling to the lattice. When $1/\tau$ is small compared to electronic relaxation rates we may formally assume that the electron-phonon coupling is 'small' and work perturbative in this coupling. This does not require the dimensionless electron-phonon coupling to be numerically small, it is the comparison to purely electronic timescales that matters. To obtain $1/\tau$ to leading order in this 'small' electron-phonon coupling, then, the Green's function $\operatorname{Im} G^R_{\dot{E}\dot{E}}(\omega)$ itself can be computed in the theory with vanishing electron-phonon interaction. The memory function approach is designed to bring out this simplification, and has been successfully used to describe slow momentum relaxation due to weak translation symmetry breaking [17–19] as well as slow phase relaxation due to dilute vortices in superfluid flow [20, 21]. Here we are extending the method to slow energy relaxation due to weak coupling to the lattice.

To make use of the Kubo formula (8) we need to specify the dynamics of the phonons. To start with we consider low temperatures and, correspondingly, acoustic phonons. We will take the phonon Hamiltonian to be

$$H_{\text{ph}} = \sum_s \int \frac{d^d k}{(2\pi)^d} \omega_{ks} a^\dagger_{ks} a_{ks} \ . \tag{9}$$

Here $s$ labels the acoustic phonon modes, with phonon creation operators $a^\dagger_{ks}$. We do not incorporate any anharmonicity of the phonons, which are taken to be in equilibrium with the environment as shown in Fig. 1. For illustrative purposes we will take the electron-phonon interaction to have the textbook form

$$H_{\text{e-ph}} = \int \frac{d^d k}{(2\pi)^d} V_k n_{-k} \ , \tag{10}$$

here $n_k = \int d^d x e^{-ikx} n(x)$, with $n(x)$ the charge density, and the deformation potential is

$$V_k = \frac{-iC|k|}{\sqrt{2\rho_M \omega_{k\|}}} (a_{k\|} - a^\dagger_{-k\|}) \ . \tag{11}$$

We give a continuum limit derivation of this textbook formula (e.g. [22]) in Appendix A. The polarization vector appearing in the mode decomposition of the phonon displacement is taken to be odd in $k_i \to -k_i$, which leads to the relative minus sign between $a_{k\|}$ and $a^\dagger_{-k\|}$ in (11) – see (41) in the Appendix. Here $\rho_M$ is the mass density and $C$ the deformation constant. The longitudinal mode has dispersion

$$\omega^2_{k\|} = v_s^2 k^2 \ . \tag{12}$$

The transverse acoustic modes do not appear in this interaction, and we shall suppress the band index in the following. It is clear that (10) is a highly simplified model of electron-phonon interactions and (12) a simplified model of phonon dispersion. Correlated materials often have exceedingly complex phonon dynamics and this dynamics must be taken as an input to our method. Nonetheless, once (10) and (12) are upgraded to realistic expressions for some specified material, the logic of the remaining steps below will go through unchanged.

Because $E$ is conserved within the purely electronic system, so that $[H_{\text{e}}, E] = 0$, and independent of the phonon degrees of freedom, so that $[H_{\text{ph}}, E] = 0$, we have

$$\dot{E} = i[H, E] = i[H_{\text{e-ph}}, E] = i[H_{\text{e-ph}}, H_{\text{e}}] = i \int \frac{d^d k}{(2\pi)^d} V_k [n_{-k}, H_{\text{e}}] = -\int \frac{d^d k}{(2\pi)^d} V_k \dot{n}_{-k} \ . \tag{13}$$

Here we used the fact that $V_k$ does not depend on the electronic degrees of freedom and therefore commutes with $H_{\text{e}}$. We now insert (13) into the Kubo formula (8). Using standard manipulations of nonzero temperature Green's functions – given in Appendix B – we obtain[2]

$$\frac{c}{\tau} = \frac{C^2}{\rho_M} \int \frac{d^d k}{(2\pi)^d} k^2 \left[ \frac{\omega_k}{2T} \text{csch} \frac{\omega_k}{2T} \right]^2 \frac{\text{Im} \, G^R_{nn}(\omega_k, k)}{\omega_k} \ . \tag{14}$$

This expression is physically transparent: the electronic energy decay rate is obtained by integrating over all processes that can excite a phonon with momentum $k$ and energy $\omega_k$. There is a $k^2$ weighting due to the derivative coupling of the acoustic phonon to the charge density. Frequencies above $\omega_k \sim T$ are not thermally accessible are suppressed. The only electronic quantity that needs to be determined is the spectral weight $\text{Im} \, G^R_{nn}(\omega, k)$. As we have noted, this is to be evaluated in the purely electronic theory, without electron-phonon coupling. We now discuss this quantity, after first briefly generalizing to optical phonons.

---

[2]The expressions in the Appendix make clear that (14) incorporates processes that transfer energy in both directions between the electrons and the lattice. We are considering circumstances where the electrons are hotter than the lattice and therefore the net energy flow is out of the electronic subsystem.

### 2.3 Optical phonons

A non-dispersing optical phonon mode can be modelled with the Hamiltonian

$$H_{\text{opt}} = \omega_o \int \frac{d^d k}{(2\pi)^d} b_k^\dagger b_k + \frac{g}{(2\omega_o)^{1/2}} \int \frac{d^d k}{(2\pi)^d} n_{-k} \left( b_k + b_{-k}^\dagger \right). \tag{15}$$

Here $\omega_o$ is a constant, $g$ the electron-phonon coupling and the creation modes obey $[b_k, b_q^\dagger] = (2\pi)^d \delta^{(d)}(k-q)$. An entirely analogous computation to the acoustic phonon case leads to

$$\frac{c}{\tau} = g^2 \left[ \frac{\omega_o}{2T} \operatorname{csch} \frac{\omega_o}{2T} \right]^2 \int \frac{d^d k}{(2\pi)^d} \frac{\operatorname{Im} G_{nn}^R(\omega_o, k)}{\omega_o}. \tag{16}$$

This decay rate will be exponentially small unless the temperature is above the energy of the optical mode, $T \gtrsim \omega_o$.

## 3 Evaluation of the energy relaxation rate

### 3.1 Phonon emission by the Lindhard continuum

Previous computations [1, 3] have used non-interacting electronic Green's functions to calculate the energy relaxation rate. We can firstly recover those results from (14). A non-interacting spin-half fermion with dispersion relation $\omega = \epsilon_k$ has the Lindhard function[3]

$$\left. \frac{\operatorname{Im} G_{nn}^R(\omega, k)}{\omega} \right|_{\text{Lind.}} = 2\pi \int \frac{d^d q}{(2\pi)^d} \frac{n_F(\epsilon_q) - n_F(\epsilon_q + \omega)}{\omega} \delta \left( \omega - \epsilon_{q+k} + \epsilon_q \right). \tag{17}$$

Here the Fermi-Dirac distribution $n_F(\epsilon) = 1/(e^{\epsilon/T} + 1)$. We must now insert (17) into (14).

To bring out the physics in the simplest possible setting we will take a spherical Fermi surface with parabolic dispersion, so that $\epsilon_q = (q^2 - k_F^2)/(2m)$. The Fermi velocity $v_F = k_F/m$. The energy conservation delta function in (17) then has argument $v_s k - \frac{v_F}{k_F} \left( \frac{1}{2} k^2 - qk \cos \theta \right)$, with $\theta$ the angle between $q$ and $k$. At temperatures much lower than characteristic electronic energy scales, $T \ll E_F = k_F^2/(2m)$, the Fermi-Dirac functions in (17) restrict $q$ to lie close to the Fermi surface so that we may set $q \to k_F$ in the delta function. The delta function thus picks out the angle $\cos \theta = (2v_s k_F - v_F k)/(2k_F v_F)$. This is consistent so long as

$$\frac{2k_F(v_s - v_F)}{v_F} \le k \le \frac{2k_F(v_s + v_F)}{v_F}. \tag{18}$$

This is the constraint that the phonon must lie within the Lindhard particle-hole continuum, as illustrated in Fig. 2. When $v_s < v_F$ there is no constraint from the lower bound in (18).

In Appendix C we perform the standard momentum integrals to evaluate the relaxation rate (14) using the Lindhard function (17). In a conventional metal with $v_s \ll v_F$ the upper cutoff in (18) is set by $2k_F$. It is then natural to express the timescale $\tau$ in terms of the Bloch-Grüneisen temperature $T_{\text{BG}} \equiv 2v_s k_F$ and the dimensionless electron-phonon coupling

$$\lambda \equiv \frac{S_{d-1}}{(2\pi)^d} \frac{C^2}{\rho_M} \frac{k_F^{d-1}}{v_s^2 v_F}. \tag{19}$$

---

[3]The unscreened charge dynamics must be used to compute the energy relaxation rate into phonons. The effects of screening by Coulomb interactions have already been incorporated into the electron-phonon interaction — indeed, screening is essential in order to obtain a short range electron-phonon interaction in the first place. The electron-phonon interaction therefore expresses the coupling of the electrons to the total electric field created by lattice vibrations, including screening. The electronic response to the total electric field is then given by the unscreened Green's function, see e.g. §7.4.2 of [22].

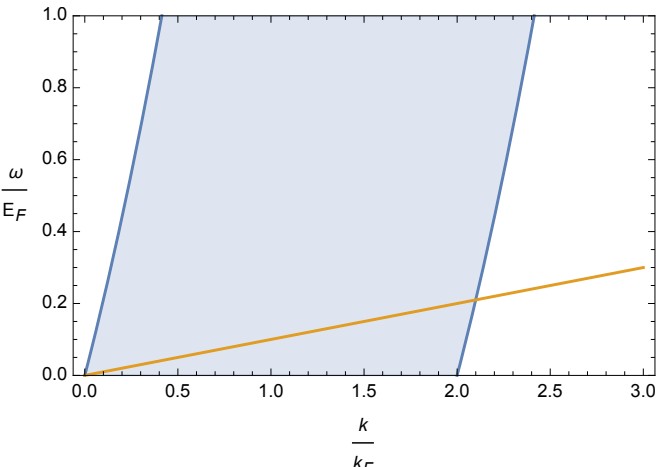

Figure 2: The shaded blue region is the phase space of Lindhard particle-hole pairs. The orange line is the dispersion of an acoustic phonon with $v_s = \frac{1}{20} v_F$. The constraint (18) is obeyed where the phonon is inside the particle-hole continuum.

This normalization agrees with e.g. [22]. In three dimensions, for example, we obtain

$$\frac{1}{\lambda} \frac{1}{T_{\mathrm{BG}} \tau} = \frac{3}{2\pi} \left(\frac{T}{T_{\mathrm{BG}}}\right)^3 \int_0^{\frac{T_{\mathrm{BG}}}{T}} dx \, x^5 \mathrm{csch}^2 \frac{x}{2} \,, \tag{20}$$

where we used the specific heat $c = \frac{1}{3} \frac{k_F^2}{v_F} T$. In the low temperature limit the integral in (20) gives $480\zeta(5) \approx 498$ and (20) is then in exact agreement with an expression in [1]. This provides a check on our formalism. More generally, the rate (20) is plotted as a function of $T/T_{\mathrm{BG}}$ in Fig. 3. At low temperatures $1/\tau \approx 237\lambda T^3/T_{\mathrm{BG}}^2$ while at high temperatures $1/\tau \approx 0.47\lambda T_{\mathrm{BG}}^2/T$. The rate peaks at $T_\star \approx 0.3 T_{\mathrm{BG}}$ with the value of $1/\tau \approx 0.9\lambda T_{\mathrm{BG}} \approx 3\lambda T_\star$. This maximum rate approaches a 'Planckian' value once $\lambda \sim 0.3$ (cf. [23], page 54).

As a crude estimate of the timescale $\tau$, assume the dimensionless electron-phonon coupling $\lambda \sim 0.2$, a typical value for non-superconducting metals [24] and take a typical value $T \sim T_{\mathrm{BG}} \sim 300$ K. Restoring factors of $\hbar$ and $k_B$ we obtain

$$\tau \sim 0.3 \text{ ps} \,. \tag{21}$$

As is clear from Fig. 3, this timescale will get longer at both low and high temperatures.

## 3.2 Destroying and evading the Lindhard continuum

### 3.2.1 Bad metals

Interactions lead to a mean free path $\ell_e$ and lifetime $\tau_e$ for the electronic single-particle Bloch states. We will mostly be interested in strongly inelastic collisions that distribute energy as well as momentum among the particles. Such interactions broaden the single-particle states, producing a thermalized many-body medium. At low frequencies $\omega \lesssim 2\pi/\tau_e$ and momenta $k \lesssim 2\pi/\ell_e$ the charge density spectral weight is universally described by collective many-body dynamics (diffusion, in the simplest case). The Lindblad continuum will only re-emerge at larger frequencies and momenta, at which collisions have not yet occurred.

The collective regime therefore gives a universal contribution to the relaxation rate $1/\tau$ from the low momentum part of the integral in the Kubo formula (14). In a conventional metal the electronic quasiparticles have a long mean free path so that $2\pi/\ell_e \ll k_F$. This

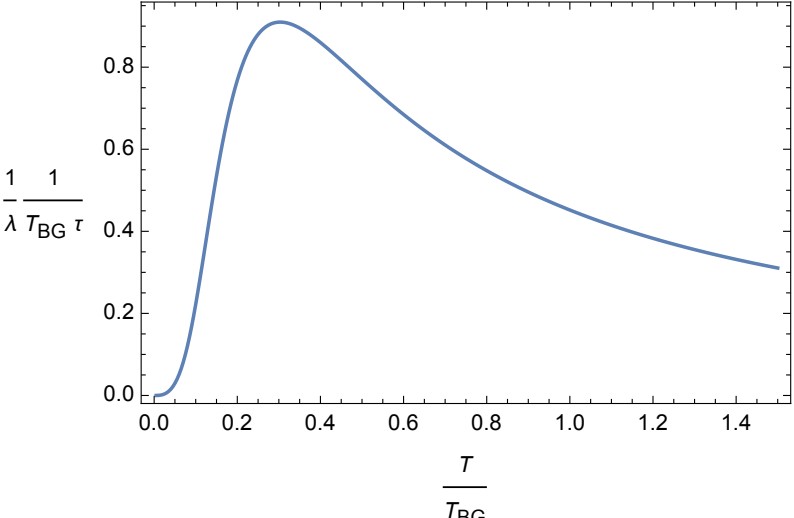

Figure 3: Rate of energy loss to phonons from the Lindhard continuum as a function of temperature in three dimensions from (20). The rate becomes small at low temperatures because there are few phonons around. The rate becomes small at high temperatures because high temperature electron-phonon scattering is elastic from the perspective of the electrons and does not efficiently transfer energy out of the electronic subsystem.

implies that only a small fraction of the dissipated modes are in the collective regime. Most dissipation instead occurs via modes in the range $2\pi/\ell_e \lesssim k \lesssim 2k_F$ that have low energy but large momentum. These modes can be approximately described by the Lindhard function (17). This fact is illustrated in Fig. 4.

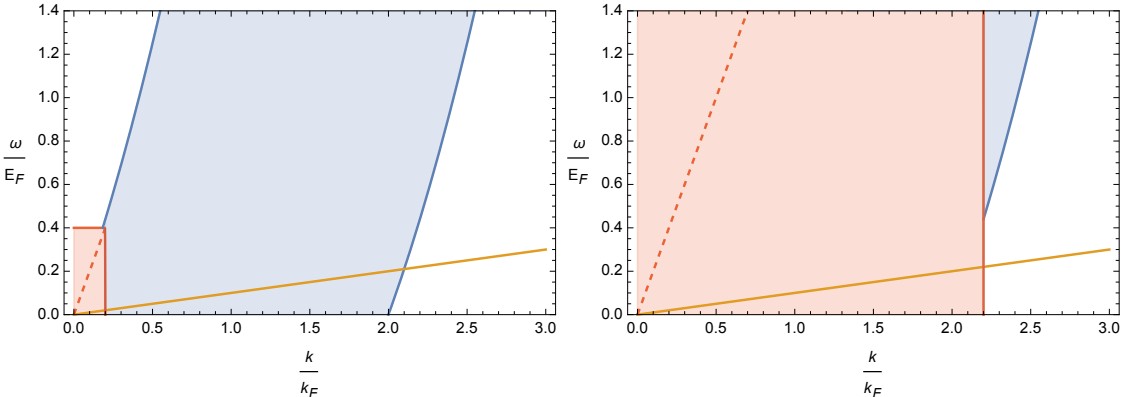

Figure 4: Collective (shaded red) and Lindhard (shaded blue) regimes in a conventional metal (left) and a bad metal (right). The collective regime extends over $k \lesssim 2\pi/\ell_e$. The good metal has $2\pi/\ell_e \ll k_F$ while the bad metal has $2\pi/\ell_e \sim 2k_F$. The dispersion of the acoustic phonon is the orange line. The dashed red line shows $\omega = v_F k$, with $\ell_e = v_F \tau_e$.

We estimate in Appendix D that in conventional metals with a long mean free path the Lindhard contribution strongly dominates the integral for $1/\tau$, as is suggested by Fig. 4. This explains why the classic treatment of electronic energy loss in conventional metals [1, 3] is justified — the emission of phonons is dominated by density fluctuations with low energy but large momentum. The Fermi surface supports a large number of such fluctuations, the

Lindhard continuum, extending up to $k = 2k_F$. These fluctuations occur on inverse momentum scales that are short compared to the mean free path and are therefore not strongly impacted by interactions.

However, if the mean free path becomes sufficiently short the collective regime 'eats up' the entire low energy Lindhard continuum. In particular, the low energy charge dynamics becomes collective over all momenta once $2\pi/\ell_e \sim 2k_F$. The relaxation rate $1/\tau$ is then fully determined by the collective dynamics in this case, rather than particle-hole pairs, as is also illustrated in Fig. 4. Metals with such short mean free paths are called bad metals [11–13]. Because the quasiparticle mean free path is likely not a well-defined concept in such regimes, it may be fruitful to *define* a bad metal as one where the low energy charge dynamics is collective (more specifically, we shall see, diffusive) at all momenta. From this perspective, bad metals show a remarkable universality — collective charge dynamics usually arises as a long wavelength effective theory [25,26], but in a bad metal it is also the microscopic theory! This universality of charge dynamics makes bad metals especially amenable to our Kubo formula for energy dissipation, as we shall explore below.

### 3.2.2 Slow metals

A second, distinct, circumstance in which the classic formula for energy relaxation will not hold is when the Fermi velocity is small, $v_F < v_s$. In this case the acoustic dispersion is outside of the Lindhard continuum at low energies $\omega \ll E_F$, as illustrated in Fig. 5 We will refer to

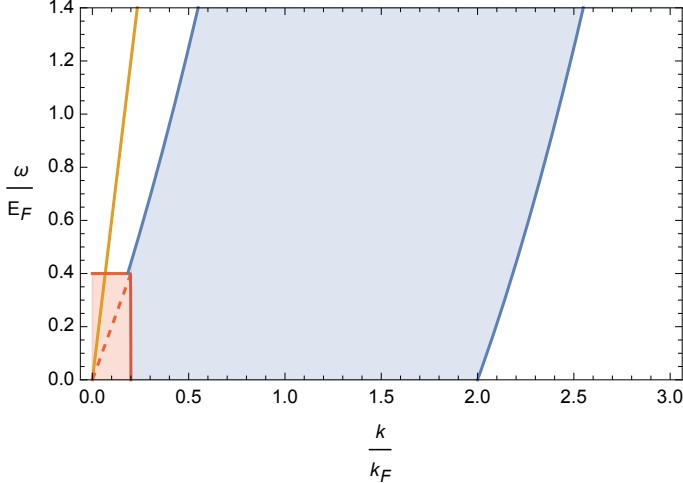

Figure 5: The orange line shows the dispersion of an acoustic phonon with $v_s = 3v_F$, outside of the Lindhard continuum (shaded blue) but inside the collective regime (shaded red).

such systems as 'slow' metals. It follows that at temperatures $T \ll E_F$ the rate of energy loss through the Lindhard continuum is exponentially small at best.

Fig. 6 illustrates why a single electron is kinematically unable to emit or absorb a phonon when $v_F < v_s$ (in the absence of inter-band scattering, cf. [27]). However, multi-electron processes are able to emit phonons. In particular, interactions will spread the charge density spectral weight down to low momenta, outside of the non-interacting Lindhard continuum. At low energies this spectral weight will have a universal collective form, as is shown in Fig. 5. More generally, when the sound speed is large compared the Fermi velocity, the relevant charge density spectral weight can be obtained in terms of the $k = 0$ optical conductivity $\sigma(\omega)$. As we will see below, our Kubo formula again gives a robust and simple expression for the decay rate $1/\tau$ in this limit.

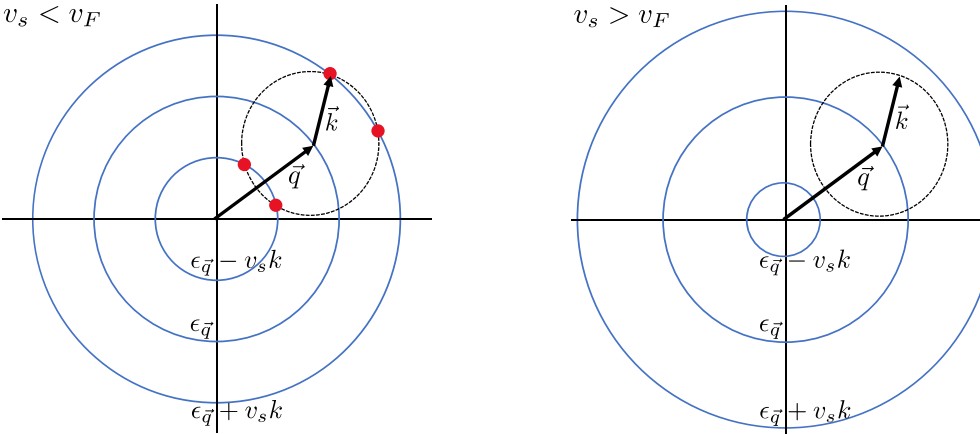

Figure 6: An electron with momentum $\vec{q}$ absorbing or emitting a phonon with momentum $\vec{k}$ acquires final momentum $\vec{q} \pm \vec{k}$ and energy $\epsilon_{\vec{q}} \pm v_s k$. These two conditions can be satisfied only if $v_s < v_F$, as in the left figure, where the four red dots represent the possible final momenta. The blue circles are constant energy contours for the electron. If $v_s > v_F$ the contours are spaced further apart relative to the phonon momentum and no final momenta satisfy momentum and energy conservation, as in the right figure.

### 3.3 Phonon emission by an incoherent spectrum

As a first explicit example of phonon emission by an incoherent spectrum, we will use the recent M-EELS measurements in optimally doped BSCCO that were mentioned in the introduction. These measurements reported an incoherent charge density spectrum without the dispersive features present in the Lindhard continuum. At $T = 295$ K and for frequencies $\omega \gtrsim 0.1$ eV [9]:

$$\operatorname{Im} G_{nn}^R(\omega, k)\big|_{\text{expt.}} = A \left( \frac{ak}{2\pi} \right)^2 \tanh \frac{\omega_c^2(k)}{\omega^2} . \tag{22}$$

The constant $A \approx 28 \times 10^{-3}/(\text{eV Å}^3)$, the lattice spacing $a = 3.81$Å, and $\omega_c(k) \approx 1$ eV, with only a weak $k$ dependence.

The conventional formula for energy relaxation — based on the Lindhard function — is manifestly inapplicable in this system. The dissipation of energy into a high energy optical phonon mode with frequency above 0.1 eV can instead be obtained by using the experimentally determined $\operatorname{Im} G_{nn}^R(\omega, k)$ from (22) directly in our Kubo formula (16). Because the temperature at which (22) was measured is relatively small compared to these energies, decay into such a mode will be Boltzmann suppressed. Presumably emission into lower energy phonon modes will be dominant in practice, but those processes are not described by (22) which is restricted to $\omega \gtrsim 0.1$ eV. We will use (22) to explicitly compute the decay rate into an $\omega_o \approx 0.1$ eV phonon, to illustrate the use of our formula. From (16) we obtain

$$\frac{1}{\tau} \approx \frac{A g^2}{c} \left[ \frac{\omega_o}{2T} \operatorname{csch} \frac{\omega_o}{2T} \right]^2 \frac{1}{\omega_o} \tanh \frac{\omega_c^2}{\omega_o^2} \frac{1}{\Delta z} \int \frac{d^2 k}{(2\pi)^2} \left( \frac{ak}{2\pi} \right)^2 \tag{23}$$

$$\sim 10 \, \text{ps}^{-1} . \tag{24}$$

In the first line $\Delta z \approx 15$Å is the interlayer spacing. To get the (very rough) final estimate we integrated $k_x$ and $k_y$ from $-\pi/a$ to $\pi/a$, set $g^2 = \omega_o^3 a^2 \Delta z$ to define an order one dimensionless electron-phonon coupling (in three dimensions) and estimated the three dimensional electronic specific heat $c \sim 0.5/(a^2 \Delta z)$ at $T = 295$ K. This crude estimate is obtained from the

fact that several other cuprates have electronic specific heat $c/T \sim 1$ mJ/(g-at. K$^2$) [28]. Despite the Boltzmann suppression and the incoherence of the spectrum, the energy dissipation time $\tau$ into this mode is seen to be faster than our previous estimate (21) for a conventional metal at the same temperature (and lower phonon frequency). The timescale for the excitation of a high energy optical phonon may be observable in a mode-resolved pump-probe experiment and is a prediction of the spectral function measured by M-EELS.

### 3.4  Phonon emission and diffusion in the high $T$ Hubbard model

We have explained above that once the mean free path becomes of order $1/(2k_F)$ the low energy Lindhard continuum is fully dissolved into collective charge dynamics. If the collisions are inelastic and thermalize the system, the charge dynamics at $\omega \lesssim 2\pi/\tau_e$ and $k \lesssim 2\pi/\ell_e$ is strongly constrained due to conservation of charge and spatial locality of the microscopic interactions. If we furthermore assume that the electronic scattering degrades the total momentum, and that no symmetries are spontaneously broken, then this collective dynamics will be diffusive.[4] The simplest model spectral weight that describes the onset of diffusion beyond a timescale $\tau_e$ is [15]

$$\left.\frac{\operatorname{Im} G_{nn}^R(\omega,k)}{\omega}\right|_{\text{dif.}} = \frac{\chi D k^2}{\omega^2 + (Dk^2 - \tau_e \omega^2)^2}. \tag{25}$$

Here $\chi$ is the charge susceptibility and $D$ the diffusivity. At low frequencies $\omega\tau_e \ll 1$ the form (25) becomes purely diffusive. At low momenta $Dk^2/\omega \ll 1$ it instead acquires a form that is consistent with a Drude optical conductivity:

$$\sigma(\omega) = \lim_{k\to 0}\frac{\operatorname{Im} G_{jj}^R(\omega,k)}{\omega} = \lim_{k\to 0}\frac{\omega^2}{k^2}\frac{\operatorname{Im} G_{nn}^R(\omega,k)}{\omega} = \frac{\chi D}{1 + \tau_e^2 \omega^2}. \tag{26}$$

As always, the dc conductivity $\sigma_{\text{dc}} = \chi D$.

Remarkably, but consistent with our general expectations outlined above, the form (25) has recently been shown to control the charge dynamics of the high temperature Hubbard model as simulated in an ultracold atomic lattice [14]. Furthermore, recent Monte Carlo computations of the spectral weight $\operatorname{Im} G_{nn}^R(\omega,k)$ in [29–31] also appear to be compatible with the form (25) over a substantial portion of the Brillouin zone, although this was not remarked upon. Fig. 7 shows $\operatorname{Im} G_{nn}^R(\omega,k)$ according to (25) as a function of $\omega$ and $k$, which can be compared to Fig. 8 in [31] between the $\Gamma$ and $K$ points. While the two dimensional Lindhard function is similarly peaked along a linearly dispersing curve, its peak is highly asymmetric in $\omega$, vanishing for $\omega > v_F k_F$. In contrast the peaks seen in [29–31] are Lorentzian as a function of $\omega$, as in (25). While this comparison is promising, the numerical lineshapes in [29–31] are obtained using analytic continuation from imaginary time data and therefore may not be accurate. Future high resolution numerical or experimental study of $\operatorname{Im} G_{nn}^R(\omega,k)$ in the Hubbard model strange metal is desirable to corroborate the diffusive model that we develop here. Finally, once $\tau_e \omega \gtrsim 1$ it is natural for $D$ to become $k$ dependent, so that the peak follows the electronic dispersion, as is seen in [31]. The simplification of a constant $D$ and $\tau_e$ in (25) will not affect the estimates below.

In Appendix C we evaluate the Kubo formula (14) on the diffusive spectral function (25). We consider an incoherent regime where the form (25) holds up to the Brillouin zone cutoff $k \sim \pi/a$. The resulting expression is simplified by estimating the relative importance of various terms. Assuming that $v_F \gg v_s$ and restricting to temperatures $T \gtrsim T_D \equiv \pi v_s/a$ we obtain, in

---

[4]With conserved momentum, charge dynamics is described by electronic sound waves instead of diffusion.

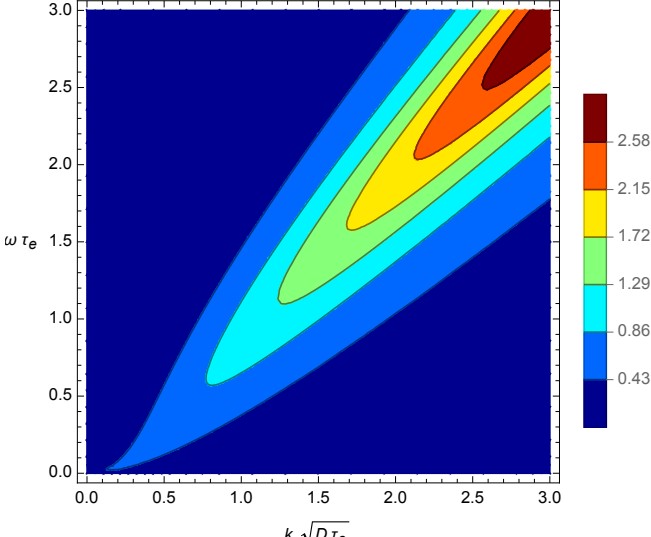

Figure 7: Contour plot of $\frac{1}{\chi}\mathrm{Im}\,G^R_{nn}(\omega, k)$ from (25) as a function of dimensionless frequency and momentum.

$d = 2$,

$$\frac{1}{\tau} = \frac{C^2}{\rho_M}\frac{\pi}{4a^2}\frac{\chi}{c}\frac{1}{D}. \tag{27}$$

Note that $\tau_e$ does not appear in this expression. This is because, due to small $v_s$, the $\omega$ terms in (25) are subleading along the phonon dispersion.

The electronic quantities $\chi$ and $D$ appearing in (27) have been measured in a simulation of the two dimensional Hubbard model in an ultracold atomic system [14] — where precisely the diffusive form (25) was used to fit the data. The model was simulated with $U \approx 7.5t$ and near-critical density $n \approx 0.82/a^2$ (with $a$ the lattice spacing) over the temperature range $0.3 \lesssim T/t \lesssim 8$. Over this range it was found that

$$D \approx ta^2\left(\frac{5}{1 + T/t} + 0.7\right), \qquad \chi \approx \frac{1}{ta^2}\frac{1}{2}\frac{1}{3 + T/t}. \tag{28}$$

These lead to near-linear resistivity $\rho = 1/(\chi D)$ over this temperature range. These results are furthermore corroborated by quantum Monte Carlo numerics [32]. With the cuprate-like value $t = 4000$ K, for example, the lower end of the temperature range considered is $T \approx 1200$ K. As we noted in the introduction, such high electronic temperatures are created in the pump-probe experiments that measure electronic energy relaxation to the lattice.

To obtain the relaxation timescale from (27) we furthermore need the specific heat $c$. It is important to compute the specific heat at constant density, as this differs from the specific heat in the grand canonical ensemble at high temperatures where thermoelectric effects are significant (see e.g. [33], supplementary material). The specific heat was not determined in the experiment but is known from Monte Carlo, e.g. [34, 35]. The data in those papers is decently fit over the relevant temperature range by

$$c \approx \frac{1}{a^2}\frac{3.3}{T^2/t^2 - 4T/t + 16}. \tag{29}$$

Using (28) and (29) in (27) the decay rate can be written as

$$\frac{1}{T_D\tau} = \lambda'\frac{T_D}{t}F\left(\frac{T}{t}\right). \tag{30}$$

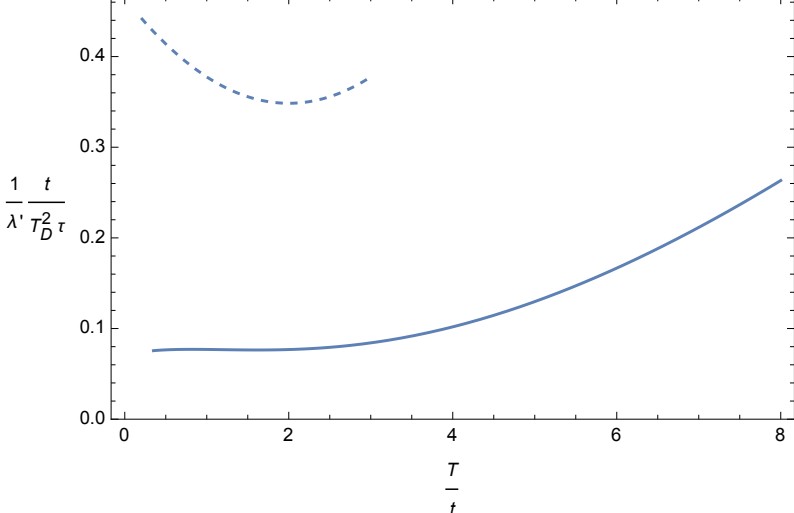

Figure 8: Solid curve: Energy relaxation rate (30) of the bad metal observed in the ultracold atom experiments of [14]. Dashed line: High temperature energy relaxation rate of a Lindhard continuum with the same parameters in the degenerate range $T < E_F$.

The function $F(T/t)$ is shown in Fig. 8. As above, the Debye scale is defined as $T_D \equiv v_s \pi/a$ and we introduced the natural dimensionless electron-phonon coupling for this regime

$$\lambda' \equiv \frac{1}{2\pi} \frac{C^2}{\rho_M} \frac{1}{v_s^2} \frac{1}{ta^2} . \tag{31}$$

This coupling is numerically very close to the previous coupling (19) if we were to use $k_F \approx 2.3/a$, that follows from the density via $n = k_F^2/(2\pi)$, and extract the velocity $v_F$ by setting $D = \frac{1}{2} v_F^2 \tau_e$. Using $D$ given above and $\tau_e$ as given in [14] leads to an approximately temperature-independent velocity $v_F \approx 3ta$. With this velocity the length $\ell_e = v_F \tau_e$ decreases from around $2a$ to $a$ over the temperature range, consistent with the bad metallic nature of this transport. Furthermore, this velocity leads to an estimate of the Fermi energy of $E_F \approx \frac{1}{2} v_F k_F \approx 3t$ which is consistent with the crossover scale of the susceptibility $\chi$ in (28) from degenerate to high temperature Curie behavior.

Fig. 8 also shows, for comparison, the corresponding relaxation rate from a two dimensional Lindhard continuum. This rate has been computed as described in Appendix C, with $k$ integrated up $\pi/a$. To make the comparison meaningful we proceed to use the values of $k_F$ and $v_F$ described in the previous paragraph, and we write the answer in terms of the coupling $\lambda'$. At high temperatures $T \gg T_D$ the formula in the Appendix gives $\frac{c}{\tau} \approx \lambda' T_D^2/t \times 0.1/a^2$. To obtain the decay rate $1/\tau$ in a way that allows us to focus on the effects of spectral incoherence, we have used the Hubbard model specific heat (29) in Fig. 8. The increase in $1/\tau$ at high temperatures, visible in both plots in Fig. 8, is due to the fact that the constant density specific heat becomes small at temperatures $T \gg t$.

Fig. 8 suggests that the absence of the Lindhard continuum in bad metals leads to slower energy dissipation into phonons, for a fixed specific heat. At $T \sim t$ the difference between the two curves is a factor of about 5. The numerical similarity between the two dissipation mechanisms is likely because the short mean free path $\ell_e \sim a \sim 1/k_F$ means that there is no small parameter in bad metals. In general interactions redistribute spectral weight, but because our formula (14) involves an integral over $k$ it is not a given whether this redistribution will cause the energy dissipation rate to increase or decrease. As the temperature is lowered, eventually the mean free path will grow long enough, $\ell_e \gg 1/(2k_F)$, that the Lindhard continuum

will re-emerge. In Appendix D we verify that once this occurs the Lindhard contribution will rapidly swamp the small diffusive regime, as is suggested by Fig. 4 and Fig. 8, leading to faster energy relaxation.[5]

### 3.5  Phonon emission from slow metals

We explained in §3.2.2 above that when $v_F < v_s$ the acoustic phonon dispersion is outside of the Lindhard continuum. Energy dissipation must therefore go via collective charge dynamics at low momenta. The limit $v_F \ll v_s$ is especially tractable, and we will focus on this case. The phonon remains at very small momenta relative to electronic momentum scales, so that we may take the $k \to 0$ limit of $\operatorname{Im} G_{nn}^R(\omega, k)$. As we recalled in (26) above the low momentum spectral weight can be expressed in terms of the optical conductivity $\sigma(\omega)$. The Kubo formula (14) becomes

$$\frac{c}{\tau} = \frac{C^2}{\rho_M} \frac{S_{d-1}}{(2\pi)^d} \frac{1}{v_s^{d+4}} \int_0^\infty d\omega\, \omega^{d+1} \left[ \frac{\omega}{2T} \operatorname{csch} \frac{\omega}{2T} \right]^2 \sigma(\omega). \tag{32}$$

Given an experimentally measured $\sigma(\omega)$ this expression predicts the energy relaxation rate.

To estimate the magnitude and temperature dependence of the relaxation rate, we can consider a three dimensional slow metal with a Drude optical conductivity as in (26). We furthermore use the Drude formula $\sigma_{\mathrm{dc}} = n\tau_e/m$ with $n = \frac{k_F^3}{3\pi^2}$ and $m = \frac{k_F}{v_F}$. The specific heat $c = \frac{1}{3} \frac{k_F^2}{v_F} T$. We also restrict to sub-Planckian metals with $\tau_e T \gtrsim 1$, the integral in (32) may be done explicitly. We obtain

$$\frac{1}{T_{\mathrm{BG}} \tau} \approx \lambda \left( \frac{T_F}{T_{\mathrm{BG}}} \right)^3 \left( \frac{T}{T_{\mathrm{BG}}} \right)^3 \frac{671}{\tau_e T}. \tag{33}$$

Here $\lambda$ is defined in (19) and $T_{\mathrm{BG}} \equiv 2k_F v_s$ and $T_F \equiv \frac{1}{2} k_F v_F$. The limit $v_F \ll v_s$ implies the unfamiliar relation $T_F \ll T_{\mathrm{BG}}$ and hence $T_F/T_{\mathrm{BG}}$ is small in (33). In particular, at low temperatures $T \ll T_{\mathrm{BG}}$ the rate (33) is suppressed by a factor of $(T_F/T_{\mathrm{BG}})^3$ relative to the low temperature limit of a conventional metal discussed below (20). Thus we see that, at low temperatures, slow metals have significantly longer energy relaxation times $\tau$ than conventional metals with comparable phonon parameters.

## 4  Discussion

The most basic point we wish to make is that the Kubo formula (14) is a powerful and general formula for the rate of energy dissipation from an electronic system into a lattice. While we have evaluated the formula in a few situations, more generally it can be a precision tool if the phonon dispersions and dynamical charge susceptibility $\operatorname{Im} G_{nn}^R(\omega, k)$ are known.

A second important notion that we have developed is that when the electronic mean free path becomes small, one expects $\operatorname{Im} G_{nn}^R(\omega, k)$ to acquire a universal diffusive form over a significant portion of the Brillouin zone. There has been significant recent progress in constructing a systematic theory of diffusion, including nonlinearities and fluctuations [25, 26], and this machinery can now be applied to the problem of bad metals in general and their coupling to phonons in particular. It also highlights that the diffusivity is a fundamental quantity in bad metals [37].

---

[5]This crossover from Lindhard to collective charge dynamics as a function of the electronic mean free path is analogous to that occurring in sound attenuation, e.g. [36]. The difference is that energy relaxation integrates along the phonon dispersion while sound attenuation occurs at a fixed phonon frequency. This makes the Lindhard contribution to energy relaxation more resilient, as high momenta are always included in the integration.

We have furthermore introduced the notion of a 'slow' metal, with $v_F < v_s$. The most obvious candidates for slow metals are those with a very small Fermi surface. In practice, however, it is not straightforward to find examples of this kind: semi-metals typically develop linear electronic dispersion at low density so that the Fermi velocity remains large, while dilute semiconductors typically become insulating. A further basic question is whether quantum critical metals can become slow. In many theories of quantum criticality, the Fermi surface survives at the critical point while the electronic mass diverges and $v_F \to 0$. If this indeed occurs with $v_s$ remaining constant, then quantum critical metals will be in a very different kinematic regime of electron-phonon interactions than conventional metals with $v_s \ll v_F$. It may be interesting to reconsider theories of quantum criticality coupled to acoustic phonons, including the intermediate regime where $v_s \approx v_F$.

A key motivation behind our work was to clarify the role of electron-phonon interactions in strange metals. In particular, it is crucial to understand, quantitatively, the contribution of electron-phonon scattering to unconventional transport [38,39]. The transfer of energy to the lattice gives an unambiguous measure of the presence and strength of electron-phonon interactions, but making use of this fact requires a formula for energy relaxation that does not assume a conventional Lindhard continuum. In this work we have obtained such a formula. The rough estimates we have made using this formula suggest energy relaxation timescales broadly compatible with tr-ARPES data in cuprates such as [5,40] with an order one electron-phonon coupling. However, more precise knowledge of $\mathrm{Im}\, G_{nn}^R(\omega, k)$ over relevant temperature and energy scales is highly desirable.

There is a technical point related to the previous paragraph. We have treated the electron-phonon coupling perturbatively, assuming that the dominant charge dynamics was due to purely electronic interactions. This is a widely considered scenario, as many models of strange metals omit phonons entirely. However, especially at temperatures above $T_{\mathrm{BG}}$, it is also plausible that quasi-elastic phonon scattering is an important part of the charge dynamics. For example, in a conventional metal this scattering determines the resistivity. Furthermore, the backreaction of phonons on the electrons is necessary to capture spectroscopic phenomena such as kinks in the electronic dispersion and phonon peaks in the M-EELS spectra [41, 42]. To incorporate such scattering requires a slight extension of the formalism we have developed, because $\mathrm{Im}\, G_{\dot{E}\dot{E}}^R$ in (8) must now be evaluated in the theory that includes nonzero electron-phonon interactions. However, because the scattering is quasi-elastic, the energy relaxation rate remains slow compared to the timescale of the interaction and therefore it is likely that our memory-function approach continues to work.[6]

While we have focused on timescales that could be measured in pump-probe experiments, let us end by returning to our discussion in §2.1 of non-Ohmic conduction due to Joule heating. To estimate the critical current (5) in a conventional three dimensional material we can use the Drude formula $\sigma_{(0)} = ne^2 \tau_{\mathrm{tr}}/m$, the density $n = \frac{k_F^3}{3\pi^2}$, the mass $m = \frac{k_F}{v_F}$ and, for example, consider Planckian transport $\frac{1}{\tau_{\mathrm{tr}}} = \frac{k_B T}{\hbar}$. Taking a typical value for a conventional metal of $k_F = 1.5 \times 10^8 \mathrm{cm}^{-1}$ and at temperature $T = 300$ K the estimate (21) for the energy relaxation time gives

$$j_{\text{non-Ohm}} \sim 24 \, \frac{\mathrm{A}}{\mu\mathrm{m}^2}\,. \tag{34}$$

This is a large current. An even larger similar estimate is obtained from the incoherent cuprate spectrum discussed in (3.3). With a normal state resistivity of $\rho \approx (T/300\mathrm{K})\, \mu\Omega\mathrm{cm}$ for optimally doped BSCCO (e.g. [43]), we obtain $j_{\text{non-Ohm}} \sim 100 \, \mathrm{A}/\mu\mathrm{m}^2$.

---

[6]A further point here is that because elastic interactions are not thermalizing (as they do not re-distribute electronic energy) the connection between the electronic single-particle lifetime and $\tau_e$, which controls the onset of collective charge dynamics, may not be immediate in these situations.

The critical current is generally reduced at lower temperatures. For example, in a conventional three dimensional metal at $T = 3$ K, say, with other parameters as in the previous paragraph we obtain $j_{\text{non-Ohm}} \sim 5 \times 10^6$ A/cm$^2$. From the discussion around (33), the critical current is predicted to become significantly lower that this in slow metals. This may be an interesting probe of quantum critical systems.

## Acknowledgements

We thank Steve Kivelson and Brad Ramshaw for insightful discussions, and Matteo Mitrano and Ali Husain for comments on the text. This work was partially supported by Simons Investigator award #620869 (P.G. and S.A.H.), STFC consolidated grant ST/T000694/1 (S.A.H.), Department of Energy Award DE-SC0019380 (P.G.), and an Alfred P. Sloan Foundation Research Fellowship through Grant FG-2020-13615 (P.G.).

## A   Acoustic phonons

In this appendix we derive, for ease of reference, the form of the electron-phonon coupling. We work directly in the continuum limit, taking care with factors of $2\pi$. The action for the displacement $u_i(x)$ at a point $x$, with $i = 1, \ldots, d$, is

$$S = \int dt\, d^d x\, \tfrac{1}{2}\left(\rho_M \dot{u}_i^2 - 2\mu(\partial_{(i} u_{j)})^2 - \lambda(\partial_i u_i)^2\right). \tag{35}$$

Solving the equations of motion, we write the displacement as

$$u_i(x) = \sum_s \int \frac{d^d k}{(2\pi)^d} \frac{1}{\sqrt{2\rho_M \omega_{ks}}} \epsilon_i^s \left(a_{ks} e^{i(kx - \omega_{ks}t)} + a_{ks}^\dagger e^{-i(kx - \omega_{ks}t)}\right), \tag{36}$$

where we have one longitudinal mode with $\omega_{k\parallel}^2 = \frac{2\mu + \lambda}{\rho} k^2$ and $d - 1$ transverse modes with $\omega_{k\perp}^2 = \frac{\mu}{\rho} k^2$, and $\epsilon_i^s \epsilon_i^{s'} = \delta^{ss'}$. The first factor in the integrand of (36) has been determined by imposing that

$$[u_i(x), \pi_j(x')] = i\delta_{ij}\delta^{(d)}(x - x'), \qquad [a_{ks}, a_{k's'}^\dagger] = (2\pi)^d \delta_{ss'}\delta^{(d)}(k - k'). \tag{37}$$

The corresponding phonon Hamiltonian is

$$H_p = \sum_s \int \frac{d^d k}{(2\pi)^d} \omega_{ks} a_{ks}^\dagger a_{ks}. \tag{38}$$

The electron-phonon interaction is

$$H_{\text{ep}} = \int d^d x\, n(x) V_{\text{ep}}(x), \tag{39}$$

where $n(x) = \sum_\sigma c_\sigma^\dagger(x) c_\sigma(x)$ is the electron density, and $V_{\text{ep}}$ is the deformation potential

$$V_{\text{ep}}(x) = \sum_c (V(x_i - (R_i^c + u_i^c)) - V(x_i - R_i^c)) \approx -\sum_a u_i \partial_i V(x - R^c)$$

$$\approx -\int \frac{d^d y}{a^d} \partial_i V(x - y) u_i(y) = -\int \frac{d^d y}{a^d} \frac{d^d q}{(2\pi)^d} e^{iq(x-y)} iq^i V(q) u_i(y), \tag{40}$$

where $R_i^a$ labels the equilibrium position of ion $c$, and $a$ is the lattice step. Use

$$\int d^d y\, e^{-iqy} u_i(y) = \sum_s \frac{1}{\sqrt{2\rho_M \omega_{qs}}} \epsilon_i^s (a_{qs} - a_{-qs}^\dagger)\,, \tag{41}$$

where we used $\epsilon_i^s(-q) = -\epsilon_i^s(q)$. We have

$$V_{\rm ep}(x) = -i \sum_s \int \frac{d^d q}{(2\pi)^d} \frac{V(q)}{a^d} e^{iqx} \frac{q^i \epsilon_i^s}{\sqrt{2\rho_M \omega_{qs}}} (a_{qs} - a_{-qs}^\dagger)\,. \tag{42}$$

Then, assuming that in the long-wavelength the potential becomes a constant $V(q)/a^d \to C$, we find

$$H_{\rm ep} = -iC \int \frac{d^d q}{(2\pi)^d} n(q) \frac{|q|}{\sqrt{2\rho_M \omega_{q\parallel}}} (a_{q\parallel} - a_{-q\parallel}^\dagger)\,. \tag{43}$$

## B  Green's function manipulations

Using (13) in (8) gives (it is most transparent to start off in real space)

$$\begin{aligned}
G_{\dot{E}\dot{E}}^R(t) = -\frac{i}{2}\theta(t) \int d^d x\, (\langle \{V(t,x), V(0,0)\}\rangle \langle [\dot{n}(t,x), \dot{n}(0,0)]\rangle \\
+ \langle [V(t,x), V(0,0)]\rangle \langle \{\dot{n}(t,x), \dot{n}(0,0)\}\rangle)\,,
\end{aligned} \tag{44}$$

and, in frequency domain,

$$G_{\dot{E}\dot{E}}^R(\omega) = \int \frac{d^d q\, d\omega'}{(2\pi)^{d+1}} \left( G_{VV}^S(\omega', q) G_{\dot{n}\dot{n}}^R(\omega-\omega', -q) + G_{VV}^R(\omega', q) G_{\dot{n}\dot{n}}^S(\omega-\omega', -q) \right)\,. \tag{45}$$

Note that, integrating by parts in time, (45) is symmetric between $V$ and $n$. It captures processes that transfer energy in both directions: from the electrons to the lattice and from the lattice to the electrons. The overall direction of energy flow is determined by which of the equilibrated subsystems is at greater temperature. To evaluate the Green's function $G_{VV}^R$ it is useful to start with

$$\begin{aligned}
G_{VV}^R(t,q) &= -i\theta(t) \int d^d x\, e^{iqx} \langle [V(x,t), V(0,0)]\rangle \\
&= i\theta(t) \frac{C^2}{2\rho_M} \int d^d x\, e^{iqx} \frac{d^d q_1}{(2\pi)^d} \frac{d^d q_2}{(2\pi)^d} \frac{|q_1||q_2|}{\sqrt{\omega_{q_1}\omega_{q_2}}} e^{iq_1 x} \langle [a_{q_1}(t) - a_{-q_1}^\dagger(t), a_{q_2} - a_{-q_2}^\dagger]\rangle \\
&= i\theta(t) \frac{C^2 |q|}{2\rho_M \sqrt{\omega_q}} \int \frac{d^d q_2}{(2\pi)^d} \frac{|q_2|}{\sqrt{\omega_{q_2}}} \langle [a_{-q}(t) - a_q^\dagger(t), a_{q_2} - a_{-q_2}^\dagger]\rangle\,.
\end{aligned} \tag{46}$$

To transform to frequency space we can use

$$\langle a_q^\dagger(t) a_{q'}(0)\rangle = (2\pi)^d \delta^{(d)}(q-q') b(\omega_q) e^{i\omega_q t}\,, \quad \langle a_q(t) a_{q'}^\dagger(0)\rangle = (2\pi)^d \delta^{(d)}(q-q')(1+b(\omega_q)) e^{-i\omega_q t}\,, \tag{47}$$

where $b(\omega) = 1/(e^{\omega/T} - 1)$. Then

$$\langle [a_{-q}(t) - a_q^\dagger(t), a_{q_2} - a_{-q_2}^\dagger]\rangle = (2\pi)^d \delta^{(d)}(q-q_2)(e^{i\omega_q t} - e^{-i\omega_q t}) = -2\langle [X_q(t), X_{q_2}]\rangle\,, \tag{48}$$

where $X_q = \frac{1}{\sqrt{2}}(a_q^\dagger + a_q)$. This yields

$$G_{VV}^R(\omega, q) = \int dt\, e^{i\omega t} G_{VV}^R(t, q) = \frac{C^2 q^2}{\rho_M \omega_q} G_{XX}^R(\omega, q)\,, \tag{49}$$

where

$$G^R_{XX}(\omega, q) = \int dt e^{i\omega t} \int \frac{d^d q_2}{(2\pi)^d} \left( -i\theta(t)\langle [X_q(t), X_{q_2}]\rangle \right)$$
$$= \frac{1}{2}\frac{1}{\omega - \omega_q + i\epsilon} - \frac{1}{2}\frac{1}{\omega + \omega_q + i\epsilon}.$$
(50)

In (49), the minus sign in the definition of retarded Green's function in (50) canceled with the minus sign in (48). Similarly,

$$G^S_{VV}(\omega, q) = \frac{C^2 q^2}{\rho_M \omega_q} G^S_{XX}(\omega, q),$$
(51)

with

$$G^S_{XX}(\omega, q) = \frac{\pi}{2}\coth\frac{\omega}{2T}\left( \delta(\omega - \omega_q) - \delta(\omega + \omega_q)\right).$$
(52)

These are the Green's functions for non-interacting phonons and will, of course, change if phonon anharmonicity becomes important.

Using the Green's functions we obtain

$$G^R_{\dot{E}\dot{E}}(\omega) = \int \frac{d^d q\, d\omega'}{(2\pi)^{d+1}}\frac{C^2 q^2}{\rho_M \omega_q}\left( G^S_{XX}(\omega - \omega', q)G^R_{\dot{n}\dot{n}}(\omega', q) + G^R_{XX}(\omega - \omega', q)G^S_{\dot{n}\dot{n}}(\omega', q)\right),$$
(53)

where we used that correlation functions are invariant under $q \to -q$. Using

$$\mathrm{Im}\, G^R_{\dot{n}\dot{n}}(\omega, q) = \omega^2 \mathrm{Im}\, G^R_{nn}(\omega, q),$$
(54)

we then find

$$\mathrm{Im}\, G^R_{\dot{E}\dot{E}}(\omega) =$$
$$\int \frac{d^d q\, d\omega'}{(2\pi)^{d+1}}\frac{C^2 q^2 (\omega')^2}{\rho_M \omega_q}\left( G^S_{XX}(\omega - \omega', q)\mathrm{Im}\, G^R_{nn}(\omega', q) + \mathrm{Im}\, G^R_{XX}(\omega - \omega', q)G^S_{nn}(\omega', q)\right).$$
(55)

Perform the frequency integral:

$$\int \frac{d\omega'}{2\pi}(\omega')^2 \left( G^S_{XX}(\omega - \omega', q)\mathrm{Im}\, G^R_{nn}(\omega', q) + \mathrm{Im}\, G^R_{XX}(\omega - \omega', q)G^S_{nn}(\omega', q)\right)$$
$$= \frac{\omega_q^2 \omega}{4T \sinh^2 \frac{\omega_q}{2T}}\mathrm{Im}\, G^R_{nn}(\omega_q, q),$$
(56)

where we took the leading term in $\omega \to 0$. We are left with the integral

$$\frac{\mathrm{Im}\, G^R_{\dot{E}\dot{E}}(\omega)}{\omega} = \int \frac{d^d q}{(2\pi)^d}\frac{C^2 q^2}{\rho_M \omega_q}\frac{\omega_q^2}{4T \sinh^2 \frac{\omega_q}{2T}}\mathrm{Im}\, G^R_{nn}(\omega_q, q)$$
$$= \frac{TC^2}{\rho_M}\int \frac{d^d q}{(2\pi)^d}q^2 \left[ \frac{\omega_q}{2T \sinh \frac{\omega_q}{2T}}\right]^2 \mathrm{Im}\,\frac{G^R_{nn}(\omega_q, q)}{\omega_q}.$$
(57)

This gives equation (14) in the main text.

# C  Integrals

## C.1  Lindhard integral

The integrals that arise from using (17) in (14) can be performed in two steps. Firstly, the $q$ integrals can be split into the radial direction, where $q = k_F + q_\perp$, and the remaining angular directions along the Fermi surface. The angular integrals simply give a factor of the area of the Fermi surface. At $T \ll E_F$ the dispersion can be linearized in the Fermi-Dirac functions so that $\epsilon_q \approx v_F q_\perp$. The $q_\perp$ integral is then $\int_{-\infty}^{\infty} dq_\perp [n_F(\epsilon_q) - n_F(\epsilon_q + \omega)]/\omega = 1/v_F$. Secondly, the $k$ integration measure can be written as $d^d k = k^{d-1} dk (\sin\theta)^{d-2} d\theta S_{d-2}$, where $S_{d-2}$ is the area of a $d-2$ dimensional sphere. The angle $\theta$ is determined by the energy conservation delta function, as described above (18), so that only the $k$ integral now remains, leading to

$$\left.\frac{c}{\tau}\right|_{\text{Lind.}} = \frac{C^2}{\rho_M} \frac{S_{d-1} S_{d-2}}{4(2\pi)^{2d-1}} \frac{k_F^{d-1}}{v_F^2} \left(\frac{T}{v_s}\right)^{d+1} \int dx [\sin\theta(x)]^{(d-3)} x^{d+2} \text{csch}^2 \frac{x}{2}. \tag{58}$$

Here $x = v_s k/T$ is subject to the constraint (18) as well as any cutoff at the Brillouin zone boundary $k \sim \pi/a$.

In particular, in $d = 2$ we obtain, integrating up to $k = \pi/a$,

$$\left.\frac{c}{\tau}\right|_{\text{Lind.}} = \frac{C^2}{\rho_M} \frac{1}{8\pi^2} \frac{k_F}{v_F^2} \left(\frac{T}{v_s}\right)^3 \int_0^{T_D/T} dx \frac{x^4 \text{csch}^2 \frac{x}{2}}{\sqrt{1 - \left(\frac{v_s}{v_F} - \frac{Tx}{T_{\text{BG}}}\right)^2}}. \tag{59}$$

Here, as in the main text, $T_D \equiv v_s \pi/a$ and $T_{\text{BG}} \equiv 2v_s k_F$. We are considering the case where where $T_D < T_{\text{BG}}$. If we furthermore impose $v_s \ll v_F$ and $T \gg T_D$ (as this will be the limit considered in Fig. 8) then the expression is simplified to

$$\left.\frac{c}{\tau}\right|_{\text{Lind.}} = \frac{C^2}{\rho_M} \frac{1}{2\pi^2} \frac{k_F}{v_F^2} \left(\frac{T_{\text{BG}}}{v_s}\right)^3 \int_0^{T_D/T_{\text{BG}}} dy \frac{y^2}{\sqrt{1 - y^2}}. \tag{60}$$

## C.2  Diffusive integral

Substituting the diffusive form (25) into the Kubo formula (14) gives

$$\left.\frac{c}{\tau}\right|_{\text{diff.}} = \frac{C^2}{\rho_M} \frac{S_{d-1}}{4(2\pi)^d} \frac{\chi D}{T^2} \left(\frac{T}{v_s}\right)^{d+4} \int dx \frac{x^{d+3} \text{csch}^2 \frac{x}{2}}{1 + \left(DT/v_s^2 - \tau_e T\right)^2 x^2}. \tag{61}$$

As previously we set $x = v_s k/T$. The integral runs over the Brillouin zone, so that the upper cutoff is $k \sim \pi/a$.

A few simplifications can be made in (61) with physical assumptions about the values of the parameters. At the upper cutoff $x \sim \pi v_s/(aT) \sim T_D/T$, where the Debye temperature is defined to be $T_D \equiv \pi v_s/a$. If we restrict to bad metals with $T \gtrsim T_D$, as is typically the case, the csch in (61) is well approximated by expanding at small $x$: $\text{csch}^2 \frac{x}{2} \to 4/x^2$. Furthermore, the second term in the denominator in (61) dominates over the first term, as we can see proceeding in two steps. Firstly, if we estimate $D \approx \frac{1}{d} v_F^2 \tau_e$ then the first term in the brackets dominates when $v_F \gg v_s$. In essence this is just the assumption that the characteristic velocity controlling diffusion is electronic. We can therefore neglect the $\tau_e T$ term. Secondly, at the upper endpoint of the integral the remaining term $D^2 T^2 x^2/v_s^4 \sim D^2/(v_s^2 a^2) \sim v_F^2/v_s^2 \times \ell_e^2/a^2 \gg 1$. In the final step we again used the fact that $v_F \gg v_s$, as well as the fact that $\ell_e/a$ is order one but not small in the bad metals we will consider. We also estimated $D \approx \frac{1}{d} v_F \ell_e$. It can be checked that the

dominance of the second term in the denominator at the endpoint of the integral is a sufficient condition to neglect the first term throughout the integral. We thereby obtain

$$\frac{c}{\tau}\bigg|_{\text{dif.}} = \frac{C^2}{\rho_M} \frac{S_{d-1}}{(2\pi)^d} \frac{\chi}{dD} \left(\frac{\pi}{a}\right)^d . \tag{62}$$

The diffusivity $D$ is still general in (62), the form mentioned in the paragraph above was used only to estimate the relative importance of the two terms in the denominator of (61).

## D  The diffusive regime in conventional metals

Conventional metals with a long mean free path will also have a diffusive regime at low energies. As we have discussed around Fig. 4 this regime is small once the mean free path is large compared to $2k_F$. In this Appendix we estimate the contribution of the diffusive regime to energy relaxation in a conventional metal and verify that it is small compared to the Lindhard contribution.

The evaluation of the diffusive contribution in Appendix C.2 goes through for a conventional metal. The only difference is that the integral must be cut off at $k \sim 2\pi/\ell_{\text{e}}$ instead of $k \sim \pi/a$. This is because for momenta beyond $2\pi/\ell_{\text{e}}$ the spectral function will cross over to the Lindhard form rather than being diffusive. Thus we obtain the diffusive contribution

$$\frac{c}{\tau}\bigg|_{\text{dif.}} = \frac{C^2}{\rho_M} \frac{\chi}{dD} \frac{S_{d-1}}{\ell_{\text{e}}^d} . \tag{63}$$

It is straightforward to compare the diffusive contribution (63) with the Lindhard contribution (58). We can estimate $D \approx \frac{1}{d} v_F \ell_{\text{e}}$ and $\chi \sim k_F^{d-1}/v_F$. At high temperatures one finds

$$1/\tau|_{\text{dif.}} \sim 1/(k_F \ell_{\text{e}})^{d+1} \times 1/\tau|_{\text{Lind.}} \ll 1/\tau|_{\text{Lind.}} , \tag{64}$$

so that the diffusive contribution is strongly suppressed once the mean free path becomes long. At low temperatures, using $\ell_{\text{e}} = v_F \tau_{\text{e}}$ and assuming that $\tau_{\text{e}} T \gtrsim 1$,

$$1/\tau|_{\text{dif.}} \sim (v_s/v_F)^{d+1} \times 1/(\tau_{\text{e}} T)^{d+1} \times 1/\tau|_{\text{Lind.}} \ll 1/\tau|_{\text{Lind.}} . \tag{65}$$

Here the suppression of the diffusive contribution is anchored in the velocity ratio $v_s \ll v_F$. The shrinking of the diffusive contribution as $1/\ell_{\text{e}}^{d+1}$ is not in general enough at low temperatures because the Lindhard contribution also becomes small, as $T^{d+1}$. If the scattering is strongly sub-Planckian, so that $\tau_{\text{e}} T \gg 1$, then the weaker scattering is sufficient. Because we are considering situations in which the scattering is predominantly inelastic and thermalizing it is reasonable to imposed the Planckian bound $\tau_{\text{e}} T \gtrsim 1$ [39]. Elastic scattering can be strongly super-Planckian but does not demarcate the boundary of a collective regime.

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
