# Peer review of "Joule heating in bad and slow metals"

_SciPost Physics, doi:SciPost Phys. 13, 095 (2022)_

## Round 2 · Referee Report · Anonymous (Referee 1) · 2022-5-16

Strengths

A novel perspective on the physics of the equilibration electron-lattice time in pump-probe experiments.

Weaknesses

This invokes a number of far reaching and unrealistic model assumptions that should be emphasized in the text.

Report

This is a valuable contribution that is surely worthwhile to be published. However, there is reason to be critical regarding the way the authors pitch their results: is the central result Eq. (14) as general and powerful as the authors claim?

This paper deals with an experimental area nicknamed “ultrafast” that has a track record struggling with the interpretation of the data. This in turn has a long history: the question of principle is whether these are generically “bad experiments” in the sense that the experimental process is so complex that it can only be understood when the system itself is completely understood which is quite a challenge dealing with unconventional metals. In fact, the pump probe experiments have delivered here not much more than the “phonon bottleneck” that is subject of this paper. What is the characteristic time for the hot electron system to transfer its heat to the lattice? I perceive the argument in the paper as “how far do we have to simplify matters before we can say anything” – this is by itself very useful. In this regard it drills down deeper in these matters than any other story I have seen before.

What is going on in the paper? Up to Eq. (8) the development is completely general, and quite refreshing: at least this referee has not quite encountered it before. But then the authors have to invoke particular limits and rather severe model assumptions to progress towards Eq. (14). The most severe limitation is in the weak electron-phonon coupling limit, a requirement to proceed with the evaluation of the memory function. For instance, it is actually in the dark in e.g. cuprates how strong this coupling is, especially dealing with the highly excited states (high temperature) that play a key role. One infers immediately that one looses generality in this limit. Generically tau is limited by retardation, the fact that the typical atom mass is larger by 5-6 orders of magnitude compared to the electron mass so that it takes much more time for the lattice to start moving than for the electrons. Given the weak e-ph “bottleneck” this retardation scale has completely disappeared from the exposition.

This is followed by the most severe approximation, assuming Eq. 10 for the e-ph coupling asserting that there is only a “volume” longitudinal coupling. This is no more than a textbook toy model and quite unrealistic especially in this context. Surely, in any real solid the transversal phonons do talk with a similar strength with the transversal shearing response of the electron system. That only the longitudinal charge susceptibility enters is an artefact of the toy model! But this is in turn the tip of the iceberg. In real solids, especially the chemically complicated ones as of relevance to bad metals etcetera there are zillions of phonons, all having their highly structured momentum dependent couplings to the electrons; for the cuprates see e.g. Johnson et al, PRB 82, 064513 (2010). In fact, it is a recent affair that in conventional metals like the high Tc hydrides it can be claimed that the phonon driven Tc can be computed using DFT band structure but this already takes a huge supercomputer effort to keep track of this plethora of numbers.

Hence, one cannot possibly claim that Eq. (18) is powerful and general. But this is not to say that the paper is not interesting. I perceive it as enlightening in conceptual regards. The way it works in these oversimplified, limiting circumstances is by itself a surprise. It is just entertaining to find out that given the toy model compressional coupling Eq. (10) matters are controlled by the dynamical charge susceptibility, which is subsequently used with much effect to illustrate matters painting the contrast between conventional metals on the one hand, and the circumstances found experimentally/computationally in cuprates and Hubbard models.

My recommendation is simple: the pitch of the manuscript should be toned-down a bit, emphasizing the far reaching model assumptions as I spelled out in the above that have to be made to proceed from Eq. (8) onwards to then proceed enthusiastically that even within these confines an intriguing and novel view on the “phonon bottleneck” arises.

Finally, my eyes discern Fig. 3 and Fig. 8 both addressing the temperature dependence of tau as main results. But in both cases I continue to wonder whether more can be said than is found in the text. The conventional metal result in Fig. 3 does show a pronounced peak in the rate that I have not seen before. Has this peak ever been observed in any experiment? I suspect that there is a logic behind this peak of a general qualitative nature that goes beyond what is found in the text – it may be worthwhile to give this some extra thought. Similarly, Fig. 8 showing that 1/tau is smaller in the bad metal than in the Fermi liquid took me by surprise. Beforehand I had expected it to be the other way around. What I get from the text is “just a conspiracy of numbers”. But yet again, I am wondering whether there may be a more general reasoning underlying this surprise. Again, a stimulus to the authors to give this extra thought.

Requested changes

  1. Modify the introduction in the guise of the above report.
  2. Clarify the origin of the peak in Fig. 3 in more qualitative and general terms, if possible.
  3. Clarify the surprising highlighted in Fig. 8, that \tau is longer in a "bad" metal compared to a conventional Fermi-liquid.

  • validity: high
  • significance: high
  • originality: top
  • clarity: top
  • formatting: perfect
  • grammar: perfect

Author:  Paolo Glorioso  on 2022-09-01  [id 2781]

(in reply to Report 1 on 2022-05-16)

The reply to the referee report is included in the attachment.

Attachment:

report1.pdf

---

## Round 2 · Referee Report · Anonymous (Referee 2) · 2022-6-6

Report

The authors consider the problem of heat dissipation arising from electrical current flow. With some assumptions on the form of electron-phonon coupling, the authors derive a general Kubo formula relating the energy relaxation rate to the charge susceptibility integrated over the phonon dispersion. As this derivation did not assume weak electronic interactions or the presence of quasiparticles, the formula is applicable to bad metals. The results presented are novel and highly relevant to the interpretation of time-resolved spectroscopy experiments. I am in favor of publication of the manuscript although there are a few minor points listed below that the authors should first address.

  1. In the case of heat dissipation of a metal with a well-defined Lindhard continuum, it is important to keep in mind that in general for 3d metals, almost all of the spectral weight in the Lindhard continuum goes into the plasmon once the long-range Coulomb interaction is included. The Lindhard continuum remains but with very low spectral weight. This is likely to affect estimates such as (21).

  2. The authors discuss the charge susceptibility of the Hubbard model as reported in Refs 29 and 30 and note its compatibility with (25). While the comparison is not unreasonable, it should be noted that the spectra calculated in 29 and 30 were obtained via analytic continuation, and details such as whether the lineshape is asymmetric are difficult to extract from imaginary time data. Another reference is Phys. Rev. B 92, 195108 (unfortunately this reference also relies on analytic continuation).

  3. For clarity, the authors should define n_k which first appears in (10). As I understand, the implied definition is n_k = sum_r e^(-ikr) c_r^\dagger c_r. Often in literature, n_k is used for the fermion occupation i.e. n_k = c_k^\dagger c_k, which is a different quantity.

  • validity: high
  • significance: high
  • originality: top
  • clarity: top
  • formatting: perfect
  • grammar: perfect

Author:  Paolo Glorioso  on 2022-09-01  [id 2780]

(in reply to Report 2 on 2022-06-06)

The reply to the referee report is included in the attachment.

Attachment:

report2.pdf

---

## Round 3 · Referee Report · Anonymous (Referee 2) · 2022-9-1

Report

The authors have addressed all points raised in the previous round of review. I recommend the manuscript for publication.

---

## Round 3 · Referee Report · Anonymous (Referee 1) · 2022-9-1

Report

This is easy -- as I expressed in my first report, this paper is conceptually interesting and valuable. My report was intended to alert the authors that they should not overstate the quantitative significance of their findings. This is well understood by the authors the various tweaks in the resubmitted version do effectively mitigate this issue. This present version is more balanced and should be published in Scipost physics.

---

## Editorial Decision

published